# RoiSeg: An Effective Moving Object Segmentation Approach Based on Region-of-Interest with Unsupervised Learning

Zeyang Zhang [1], Zhongcai Pei [1], Zhiyong Tang [1] and Fei Gu [2],*

1 School of Automation Science and Electrical Engineering, Beihang University, Beijing 100191, China; zzy89087@buaa.edu.cn (Z.Z.); peizc@buaa.edu.cn (Z.P.); zyt_76@buaa.edu.cn (Z.T.)
2 School of Computer Science and Technology, Soochow University, Suzhou 215006, China
* Correspondence: gufei@suda.edu.cn

**Abstract:** Traditional video object segmentation often has low detection speed and inaccurate results due to the jitter caused by the pan-and-tilt or hand-held devices. Deep neural network (DNN) has been widely adopted to address these problems; however, it relies on a large number of annotated data and high-performance computing units. Therefore, DNN is not suitable for some special scenarios (e.g., no prior knowledge or powerful computing ability). In this paper, we propose RoiSeg, an effective moving object segmentation approach based on Region-of-Interest (ROI), which utilizes unsupervised learning method to achieve automatic segmentation of moving objects. Specifically, we first hypothesize that the central n × n pixels of images act as the ROI to represent the features of the segmented moving object. Second, we pool the ROI to a central point of the foreground to simplify the segmentation problem into a classification problem based on ROI. Third but not the least, we implement a trajectory-based classifier and an online updating mechanism to address the classification problem and the compensation of class imbalance, respectively. We conduct extensive experiments to evaluate the performance of RoiSeg and the experimental results demonstrate that RoiSeg is more accurate and faster compared with other segmentation algorithms. Moreover, RoiSeg not only effectively handles ambient lighting changes, fog, salt and pepper noise, but also has a good ability to deal with camera jitter and windy scenes.

**Keywords:** Region-of-Interest; moving object segmentation; unsupervised learning; classification compensation

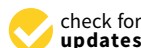



## 1. Introduction

Many researchers have proposed efficient solutions to solve foreground detection in video object segmentation problems. Among these solutions, deep neural network (DNN) methods have impressive performance with high accuracy. However, DNN architectures need enough datasets and time to train the network for improving the accuracy, which makes it not suitable for some special scenarios without enough training samples (e.g., detection of air-dropped objects in military operations) or with strict time requirements (e.g., the interception of a cannon against a shell). Moreover, these DNN-based methods also require high-performance computing units to complete all the tasks, which is too expensive for ordinary people. Background subtraction and frame difference are commonly adopted in solving the video object segmentation problems [1,2]. There are several challenges existing in background subtraction and frame difference, such as including various illumination changes, camera jitter, dynamic background, camouflage, shadows, bootstrapping and video noise [3,4]. Although many useful algorithms for background modeling have been designed, their performance is limited due to the complexity of algorithms, for example, background subtraction and the modeling of a scene based on each pixel of each frame [5]. Moreover, the accuracy of these algorithms is to some extent effected by wind noise or camera jitter [6].

To deal with these challenges, we propose RoiSeg, an effective object segmentation approach based on Region-of-interest (ROI), which utilizes unsupervised learning method to achieve automatic segmentation of moving objects. RoiSeg hypothesizes the central n∗n pixels of images as the ROI to reflect the features of moving object, then the classification of all pixels is turned into that of ROI central points. In the field of classification, the supervised learning methods usually provide a better accuracy compared with the unsupervised learning methods, however, they inevitably need more annotated datasets, hence increasing the workload of computing units [7]. To address this problem, RoiSeg adopts an automatic generation method based on ROI to produce the training samples with the unsupervised learning method. Moreover, RoiSeg also implements an online sample classifier to compensate the imbalance of different classes.

We highlight our main contributions as follows:

- We propose RoiSeg, an effective object segmentation approach based on ROI, which utilizes unsupervised learning method to achieve automatic segmentation of moving objects. RoiSeg not only effectively handles ambient lighting changes, fog, salt and pepper noise, but also has a good ability to deal with camera jitter and windy scenes.
- We hypothesize the central n∗n pixels as the ROI and simplify the foreground segmentation into a classification problem based on ROI. In addition, we propose an automatic generation method to produce the training samples and implement an online sample classifier to compensate the imbalance of different classes, respectively.
- We also conduct extensive experiments to evaluate the performance of RoiSeg and the experimental results demonstrate that RoiSeg is more accurate and faster compared with other segmentation algorithms.

The rest of this paper is organized as follows. Section 2 presents a review of related works. The description of RoiSeg is demonstrated in Section 3. The comparison experiments are given in Section 4. Finally, the conclusion is drawn in Section 5.

## 2. Related Work

Video segmentation has attracted great attention and many researchers have proposed to use DNN methods to solve this problem due to its impressive performance in this field. However, DNN is obviously not suitable for scenarios with a small/no training samples. Background subtraction, a crucial step in video object segmentation has attracted great attention in the last two decades. The main idea of background subtraction is to build a background model with a fixed number of frames. This model can be designed by different methods, such as statistical, fuzzy, neuro-inspired, and so forth. Among these methods, statistical methods have been intensively studied and widely used in various applications [8–11]. For example, Xue et al. developed a message passing algorithm termed offline denoising-based turbo message passing subtracting the background successfully with a lower mean squared error and better visual quality for both offline and online compressed video background subtraction [12]. Stauffer and Grimson implemented a parametric probabilistic background model [13]. In this model, distributions of each pixel color updated through an online expectation-minimization algorithm, were represented by a sum of weighted Gaussian distributions defined in a given color space: the Gaussian Mixture Model (GMM). Culibrk et al. adopted a neural network to determine whether each pixel of the image belongs to the foreground or the background [14]. Yu et al. established a spatio-color model based on both foreground and background, which used Expectation Maximization (EM) to track the parameters of GMM [15]. Gallego et al. used EM in the same way but modeled foreground and background at the region level and pixel level, respectively [16]. Cuevas and Garcia proposed an algorithm for foreground extraction and background updating using fuzzy functions and modeled both foreground and background in a non-parametric way [17].

These algorithms mainly implemented foreground detection on each pixel of a frame and may not be able to segment some parts of the background into foreground, resulting in a lower accuracy than DNN methods. However, these algorithms can provide some

real-time results to meet some time-sensitive tasks. In addition, it is useful to first segment an ROI by frame difference before clustering and classification. In this paper, we propose RoiSeg, an effective object segmentation method based on ROI, which utilizes unsupervised learning to improve the accuracy of foreground segmentation and ensure the real-time performance. RoiSeg includes two crucial designing methods, namely clustering and classification methods.

Cluster analysis is a statistical multivariate analysis technique which is a common method of unsupervised machine learning [18]. It divides a set of data points into several classes, with the data points in each cluster being very similar but the data points in different clusters being very different [19]. K-means is an excellent clustering method based on segmentation. It iteratively calculates the distance from each point to the K-cluster center, so that K clusters can be found in a given data set [20]. Seiffert et al. presented an efficient initial seed selection method, RDBI, to improve the performance of the K-means filtering method by locating the seed points at dense, well-separated areas of the dataset [21]. Nidheesh et al. presented an improved, density-based version of K-Means, the key idea of which is to select as the initial centroids data points which belong to dense regions and which are adequately separated in feature space [22]. The Gaussian mixed model (GMM) is a classic statistical model, in which samples are generated by a Gaussian mixture distribution and the expectation maximization (EM) algorithm is used to update the parameters of the model [23]. Unlike traditional methods of cluster analysis based on heuristic or distance-based procedures, finite mixture modeling provides a formal statistical framework on which to base the clustering procedure [24]. Theoretically, all the data points can fit as long as the GMM has enough components, but the relationship between the number of modes and the number of components in the mixture is very complex so it is particularly important to determine the number of components. In this paper, we only use two unsupervised clustering algorithms: the GMM and K-Means.

The popular Naive Bayesian classifier performs well in dealing with discrete data [25]. Naive Bayes can perform surprisingly well in classification tasks where the probability, itself calculated by the Naive Bayes classifier, is not important. In recent years, many scholars have studied Naive Bayesian classifiers and suggested several algorithms to improve their predictive accuracy [26–28]. However, classifiers trained with imbalanced data tend to generate results with a high true negative rate and low true positive rate. In data mining and machine learning, it is difficult to establish an effective classifier for imbalanced data [29]. Therefore, many scholars have proposed various methods to compensate for it [30]. The common methods are as follows: algorithmic-level methods, data-level methods, cost-sensitive methods, and ensembles of classifiers [31].

The threshold method and one-class learning method are the most efficient algorithm-level solutions; the former sets different thresholds at different learning stages for different types of data, whereas the latter uses specific data to train the classifier. Data-level solutions are based on preprocessing the collected imbalanced training data set by either downsampling or oversampling strategies. Gustavo showed that resampling solutions can effectively solve the class imbalance problem and optimize classifier performance [32]. In particular, preprocessing the imbalanced data before constructing the classier is simple and efficient because the advantage of the data-level solution is to make the sampling and classifier training processes independent [33]. The data preprocessing method is based on the resampling of imbalanced data. Oversampling approaches are used to increase the number of data samples in the minority class and downsampling approaches are used to reduce the number of data samples in the majority class, respectively [34]. Common resampling methods include the synthetic minority oversampling technique (SMOTE) and random undersampling (RUS). RUS [21] performs similarly to SMOTE, but is based on a downsampling process where some examples are removed from the majority class. Lin et al. presented a clustering-based undersampling, which uses the K-nearest to cluster the minority class into the class subset which consists of the difference between the majority class and the minority class, resulting in a balanced training set [33].

## 3. Design of RoiSeg

In this section, we will describe the designing process of RoiSeg. As shown in Figure 1, RoiSeg consists of three modules, namely, ROI-central-point generation and feature extraction, automatic training-sample generation, and an online sample classifier. The purpose of RoiSeg is to classify the foreground through the ROI central points. In the first module, the frame difference and canny edge detection are used to transform the background modeling of each pixel into an ROI-central-point-based classification problem, which greatly reduces the amount of data operation. We then extract the features of the ROI central points and provide them to the automatic training-sample generation. In the second module, the characteristics extracted from the ROI central points are made into training samples using ROI-central-point-based sample clustering and a proposed trajectory-based class classifier. In the third module, we explore the training samples and find that they are class imbalanced. A K-means oversampling method is proposed to solve the class imbalance problem and a means to update training samples online is employed to compensate for the weaknesses of the Bayesian classifier.

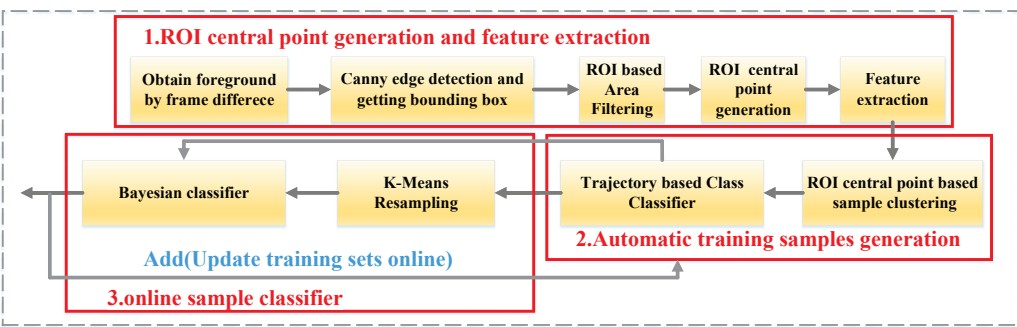

**Figure 1.** The framework of RoiSeg.

### 3.1. ROI-Central-Point Generation

We use an imbalance degree $\eta$ proposed in [35] to demonstrate the imbalance between the foreground and background, as shown in Table 1.

**Table 1.** Imbalance Degree of BMC Database And Self-captured Sequences.

| Dataset | 112 | 122 | 212 | 222 | 312 | 322 | 412 | 422 | 512 | 522 | My_video1 | My_video2 |
|---|---|---|---|---|---|---|---|---|---|---|---|---|
| $\eta$ | $\infty$ | $\infty$ | $\infty$ | $\infty$ | $\infty$ | $\infty$ | $\infty$ | $\infty$ | 0.0489 | 0.144 | 0.342 | 0.314 |

$$\eta = \frac{\text{sum}(F)}{\text{sum}(B)}, \tag{1}$$

where $\text{sum}(F)$ and $\text{sum}(B)$ are the sums of foreground and background pixels. We compute $\eta$ on several subsets of the BMC database plus two self-captured sequences ("My_video1", "My_video2"). The result reveals that the foreground and background are relatively imbalanced. $\infty$ means the number of foreground is far greater than the number of background.

As shown in Figure 2, the moving targets often include the foreground and the background [13]. The frame difference is intended to compute the difference between the current frame and the previous frame in the video sequence and then segment the moving targets. Suppose we have obtained the foreground frame shown in Figure 2c. There is a significant change in the pixel value between the current and previous frames at the position of the moving target. The moving target in the current frame is copied to the corresponding position in the previous frame, and a new previous frame is obtained. The moving target will not appear again if the new previous frame is subtracted from the current frame, as shown in Figure 3.

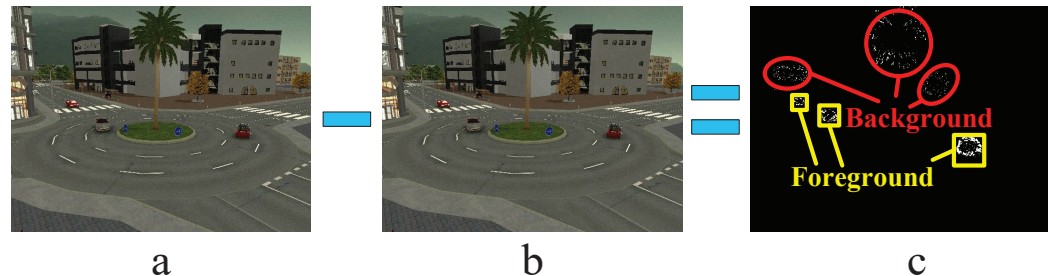

**Figure 2.** Frame difference. (**a**) Current frame. (**b**) Previous frame. (**c**) Binary frame.

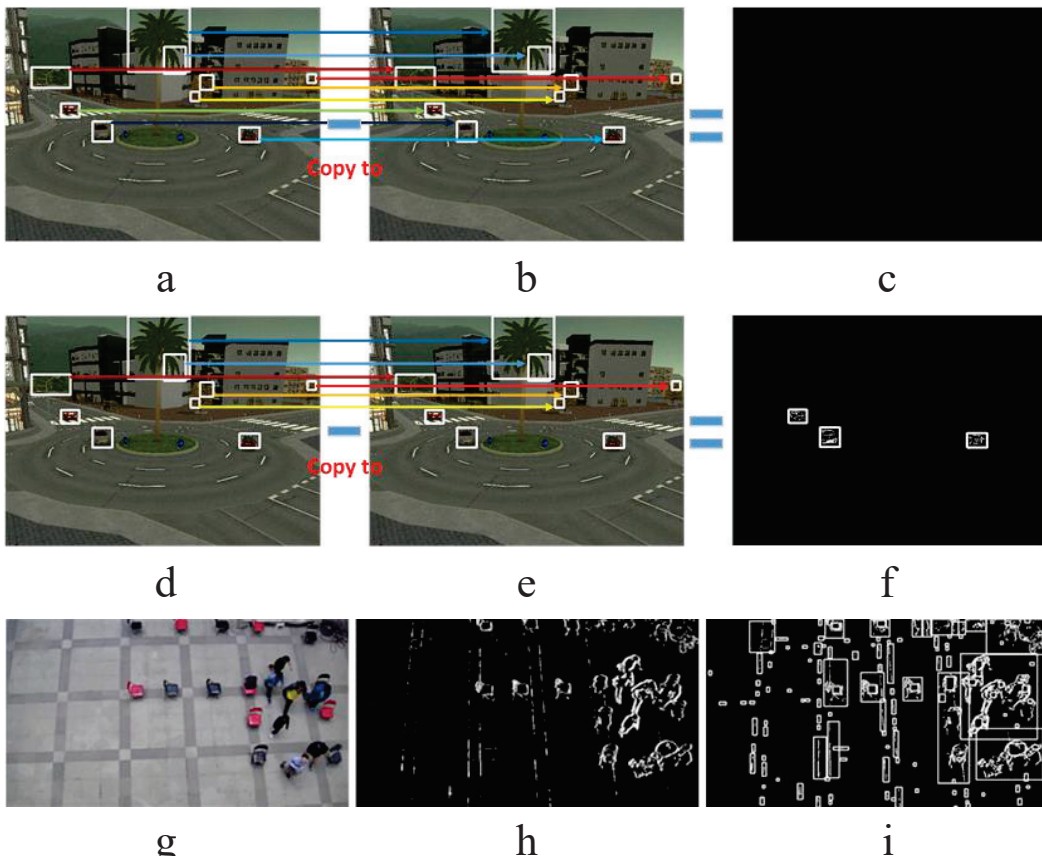

**Figure 3.** Copy all ROI in current frame to previous frame. (**a**,**d**,**g**) are current frame; (**b**,**e**) are previous frame; (**c**,**f**,**h**,**i**) are binary frame. (**a**–**c**) show the Process that copy all ROI in current frame to previous frame. (**d**–**f**) show the Process that selectively copy ROI of current frame to previous frame. (**g**–**i**) The moving target detected by the frame difference.

Following this principle, suppose we have classified the foreground and background. The background of the current frame is copied to the corresponding position in the previous frame, and a new previous frame is obtained. Foreground detection is then done if the new previous frame is subtracted from the current frame, as shown in Figure 3. When the moving target is detected by the frame difference (shown in Figure 3b), the Canny algorithm is used to detect the contours of the moving target and then the bounding boxes of the contours are obtained (as shown in Figure 3c). We call the region of the bounding boxes the Region of Interest (ROI). Therefore, the classification of foreground and background can be regarded as the classification of the bounding boxes. Furthermore, we use the center of the bounding box (the ROI central point) to represent the bounding box so that foreground detection is transformed into an ROI-central-point-based classification problem.

### 3.2. ROI-Based Noise Filter

Figure 3c shows that the areas of the bounding boxes for noise are much smaller than those of the foreground, because the bounding boxes for the vehicles and pedestrians that we pay attention to are often larger than those of other moving targets [36,37]. Based on this assumption, a bounding-box-area-based noise filter is proposed to remove the bounding boxes whose area is below a preset threshold. Here, the threshold is set to 0.1%. In this paper, we use 12 frame sequences as the experimental test set. Ten of them are from the BMC dataset [38], and two are hand-captured high-resolution crowd walking videos taken with a top-view camera, in which jitter was generated by shaking the camera. The description of the experimental test set is shown in Table 2. The filtering thresholds of the 12 videos are shown in Table 3. For the video test sets "112", "122", "212", "222", "312", "322", "412", and "422", we find that after filtering with the preset threshold, all the foregrounds are recognized, as shown in Figure 4.

**Table 2.** The description of the experimental test set.

| Sequences | Description | Size |
|---|---|---|
| 112<br>122 | Cloudy, without acquisition noise, as normal mode | 640 × 480<br>640 × 480 |
| 212<br>222 | Cloudy, with salt and pepper noise during the whole sequence | 640 × 480<br>640 × 480 |
| 312<br>322 | Sunny, with noise, generating moving cast shadows | 640 × 480<br>640 × 480 |
| 412<br>422 | Foggy, with noise, making both background and foreground hard to analyze | 640 × 480<br>640 × 480 |
| 512<br>522 | Wind, with noise, producing a moving background | 640 × 480<br>640 × 480 |
| My_video1<br>My_video2 | Camera jitter | 1280 × 720<br>1280 × 720 |

**Table 3.** Threshold of the bounding box areas to classify foreground.

| Video Sequences | 112 | 122 | 212 | 222 | 312 | 322 | 412 | 422 | 512 | 522 | My_video1 | My_video2 |
|---|---|---|---|---|---|---|---|---|---|---|---|---|
| **Number of Video clips** | 1502 | 1503 | 1499 | 1499 | 1499 | 1501 | 1499 | 1499 | 1499 | 1499 | 390 | 390 |
| **number of pixels in a bounding box** | 304 | 218 | 304 | 218 | 304 | 218 | 304 | 218 | 304 | 218 | 1000 | 1000 |
| **Total area of frame covered by bounding boxes for noise (%)** | 0.1 | 0.07 | 0.1 | 0.07 | 0.1 | 0.07 | 0.1 | 0.07 | 0.1 | 0.07 | 0.1 | 0.1 |

There are two reasons for this phenomenon: first, the frame difference has good suppression on ambient lighting changes, so the moving cast shadows and fog cannot be detected. Second, the areas of the bounding boxes caused by salt and pepper noise, and so forth, are usually far smaller than those of the foreground, such as cars and pedestrians. However, for the video test sets "512", "522", "My_video1", and "My_video2", noise produced by the wind or the camera jitter and by other dynamic background factors dominates. The areas of the bounding boxes of such noise are random, usually varying with the strength of the wind and the magnitude of the jitter. When the area of a bounding box is larger than the threshold, noise is not be removed by area-based filtering. The filtering result is shown in Figure 4. It is important to achieve high classification accuracy in the Performance Comparison for the experiment (shown in Table 4) because a lot of noise is removed. For the test sets "112", "122", "212", "222", "312", "322", "412", and "422", we successfully obtain the foreground after filtering, so we do not need to use the second and third modules to classify the foreground and the background. This is why we do not use test sets "112", "122", "212", "222", "312", "322", "412", and "422" in the experiment to classify the foreground- and background-based ROI central points and instead only use the test sets "512", "522", "My_video1", and "My_video2".

**Table 4.** Performance evaluation of the five algorithm and proposed RoiSeg.

| BMC Sequences | DPWren GABGS | | | | Mixture Of Gaussian V1BGS | | | | MultiLayer BGS | | | | Pixel Based Adaptive Segmenter | | | | LBAdaptive SOM | | | | Proposed RoiSeg | | | |
|---|---|---|---|---|---|---|---|---|---|---|---|---|---|---|---|---|---|---|---|---|---|---|---|---|
| | P | R | F | FPS | P | R | F | FPS | P | R | F | FPS | P | R | F | FPS | P | R | F | FPS | P | R | F | FPS |
| 112 | 0.87 | 0.87 | 0.87 | 70.2 | **0.96** | 0.74 | 0.84 | 89.3 | 0.92 | **0.95** | **0.93** | 5 | 0.88 | 0.9 | 0.89 | 15.6 | 0.86 | 0.92 | 0.89 | 20.6 | 0.89 | 0.93 | 0.91 | **115** |
| 122 | 0.91 | 0.87 | 0.89 | 77.6 | **0.96** | 0.7 | 0.8 | 70.5 | 0.91 | **0.94** | **0.93** | 2.2 | 0.9 | 0.88 | 0.89 | 13.2 | 0.88 | 0.93 | 0.9 | 22.3 | 0.91 | **0.94** | 0.92 | 120.6 |
| 212 | 0.92 | 0.86 | 0.89 | 58.3 | **0.97** | 0.74 | 0.84 | 70.3 | 0.94 | **0.94** | **0.94** | 2.5 | 0.89 | 0.89 | 0.89 | 8.2 | 0.79 | 0.77 | 0.78 | 15.5 | 0.89 | 0.93 | 0.91 | 70.6 |
| 222 | 0.93 | 0.86 | 0.9 | 59.2 | **0.96** | 0.7 | 0.81 | 70.6 | 0.94 | 0.93 | **0.93** | 3.5 | 0.9 | 0.87 | 0.89 | 7.6 | 0.89 | 0.92 | 0.91 | 14.2 | 0.91 | **0.94** | 0.92 | 85.1 |
| 312 | 0.65 | 0.78 | 0.71 | 70.4 | **0.98** | 0.68 | 0.8 | 73.8 | 0.96 | 0.87 | **0.91** | 2.4 | 0.88 | 0.87 | 0.87 | 11.2 | 0.52 | 0.84 | 0.64 | 19.2 | 0.89 | **0.93** | **0.91** | 103.2 |
| 322 | 0.89 | 0.78 | 0.83 | 63.2 | **0.95** | 0.65 | 0.77 | 65.9 | 0.94 | 0.85 | 0.89 | 4.3 | 0.9 | 0.8 | 0.85 | 12.3 | 0.54 | 0.85 | 0.66 | 15.1 | 0.91 | **0.94** | **0.92** | 88.3 |
| 412 | 0.53 | 0.76 | 0.62 | 62.1 | **0.98** | 0.69 | 0.81 | 87.7 | 0.71 | 0.84 | 0.77 | 3.1 | 0.85 | 0.82 | 0.84 | 11.5 | 0.51 | 0.78 | 0.61 | 13.3 | 0.89 | **0.93** | **0.91** | 98.1 |
| 422 | 0.53 | 0.75 | 0.62 | 69.3 | **0.97** | 0.64 | 0.77 | 75.8 | 0.77 | 0.79 | 0.78 | 3.9 | 0.85 | 0.77 | 0.81 | 10.4 | 0.51 | 0.78 | 0.62 | 15.1 | 0.91 | **0.94** | **0.92** | 85.8 |
| 512 | 0.63 | 0.86 | 0.73 | 73.4 | 0.82 | 0.74 | 0.78 | 76.8 | 0.65 | **0.93** | 0.76 | 4.1 | **0.82** | 0.89 | 0.86 | 14.1 | 0.52 | 0.88 | 0.66 | 18.3 | 0.81 | 0.91 | **0.86** | 102.3 |
| 522 | 0.8 | 0.86 | 0.83 | 70.3 | **0.91** | 0.69 | 0.79 | 72.2 | 0.88 | 0.93 | 0.9 | 3.3 | 0.89 | 0.87 | 0.88 | 12.4 | 0.67 | 0.92 | 0.78 | 21.6 | 0.89 | **0.93** | **0.91** | 99.1 |
| My_video1 | 0.38 | 0.84 | 0.54 | 12.1 | 0.68 | 0.52 | 0.59 | 16.3 | 0.76 | 0.89 | 0.83 | 0.5 | 0.82 | 0.89 | 0.85 | 5.1 | 0.42 | 0.8 | 0.55 | 6.5 | **0.9** | 0.91 | 0.86 | 42.23 |
| My_video1 | 0.3 | 0.83 | 0.44 | 11.3 | 0.8 | 0.48 | 0.6 | 13.5 | 0.75 | 0.84 | 0.79 | 0.45 | 0.75 | 0.85 | 0.8 | 4.5 | 0.23 | **0.87** | 0.36 | 3.2 | **0.86** | 0.86 | **0.86** | 39.62 |

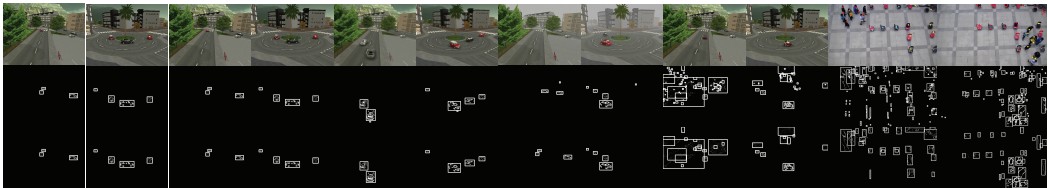

**Figure 4.** Experimental results with filtering. Left to right: "112", "122", "212", "222", "312", "322", "412", "422","512", "522", "My_video1", "My_video2". Top to bottom: Original frame, Binary frame without filtering, Binary frame with filtering.

### 3.3. Automatic Training-Sample Generation

In Section 3.2, we get input samples, but these samples are unlabeled original samples and cannot be used as training samples for the classifier. In order to obtain labeled training samples, we propose an ROI-central-point-clustering method and a class detector.

#### 3.3.1. ROI Pooling and Feature Extraction

For those videos not classified with the aforementioned noise filter, we are required to develop further feature extraction methods to enable noise reduction.

Let $Z^t = \{z_i^t\}$ be a frame at time $t$, $z_i^t$ represent each pixel in $Z^t$, and $i$ generally refers to the i-th element in the set. We choose the $(r_i, g_i, b_i)$ color feature and the coordinate $(x_i, y_i)$ of $z_i^t$ as the spatio-color feature space $z_i^t = (r_i^t, g_i^t, b_i^t, x_i^t, y_i^t)$. For the ROI central point, a 5-tuple vector in the spatio-color feature space is selected as the $z_i^t = (r_i^t, g_i^t, b_t^t, x_i^t, y_t^t)$ classification-learning feature. In Section 3.1, we used the center of the bounding box to represent the ROI central point. We also process the ROI central point with mean-pooling. Experiments show that using the pooled ROI central point $z_i^t = (r_i^t, g_i^t, b_t^t, x_i^t, y_t^t)$ as a learning feature can provide good foreground detection results, as shown in Table 4 in the columns for the proposed method. The advantage of this method is that it reduces the computation load and guarantees good classification accuracy.

#### 3.3.2. ROI Central Point Based Sample Clustering

We noticed that the background noise is often bound to a specific area. In adjacent frames, however, a moving target's coordinates are also similar. Regardless of the foreground or background, the same type of target has more similar color features R, G, and B. In this section, we cluster ROI central points with similar characteristics. A GMM is a function to estimate the probability density of the exact polymorphism. It has excellent performance in clustering and its general form is as follows:

$$P(x_i) = \sum_{j=1}^{k} \alpha_j G\left(x_i; \mu_j; \sum_j\right). \tag{2}$$

The weighted coefficient $\alpha_j$ is satisfied as follows:

$$\sum_{j=1}^{k} \alpha_j = 1, 0 \leq \alpha_j \leq 1. \tag{3}$$

The *j*-th component $(j = 1, \ldots, k)$ is shown below:

$$G\left(x_i; \mu_j; \sum_j\right) = \frac{\left|\Sigma_j\right|^{-\frac{1}{2}}}{(2\pi)^{\frac{d}{2}}} e^{-\frac{1}{2}(x_j - \mu_j)^r \Sigma_j^{-1}(x_j - \mu_j)}, \tag{4}$$

where $\mu_j$ and $\sum_j$ represent the $i^{-t_1}$ mean vector and covariance matrices, respectively. We choose $z_i^t = \left(r_i^t, g_i^t, b_i^t, x_i^t, y_t^t\right)$ as $x_i$, which is the input of the GMM. All ROI central points in the first l frames (here we set l = 5) are used as input samples for the GMM. For example, we use the 3-Component GMM, so the ROI central points are clustered into 3 similar clusters, red, blue and yellow sets, as shown in Figure 5. However, the number of components determines the clustering accuracy of GMM. We choose precision (*P*), recall (*R*), and F-Measure (*F*) for performance evaluation:

$$\text{precision} = \frac{TP}{TP + FN}, \text{ recall } = \frac{TP}{TP + FP}, \tag{5}$$

$$F = \frac{2 \times \text{precision} \times \text{recall}}{\text{precision } + \text{recall}}. \tag{6}$$

$TP, FN$ and $FP$ are the number of true positive, false negative, and false positive pixels, respectively. Figure 6 shows that *F* increases with the number of components increases. However, when this number reaches 6, the growth of F tends to be slow. Figure 7 shows that FPS drops with the increase of the number of components. The reasons are as follows: with the number of GMM components increasing, the data-fitting ability of the GMM is gradually enhanced, so the *F* value increases; meanwhile, the computation load is also increasing, which leads to the reduction of FPS; finally, the number of components increasing makes it easy for the GMM to over-fit the data. As such, we set the number of components to 6. Figure 6 demonstrated that F is between 0.87 and 0.91, which means that we can use the pooled ROI center point as the input of the GMM.

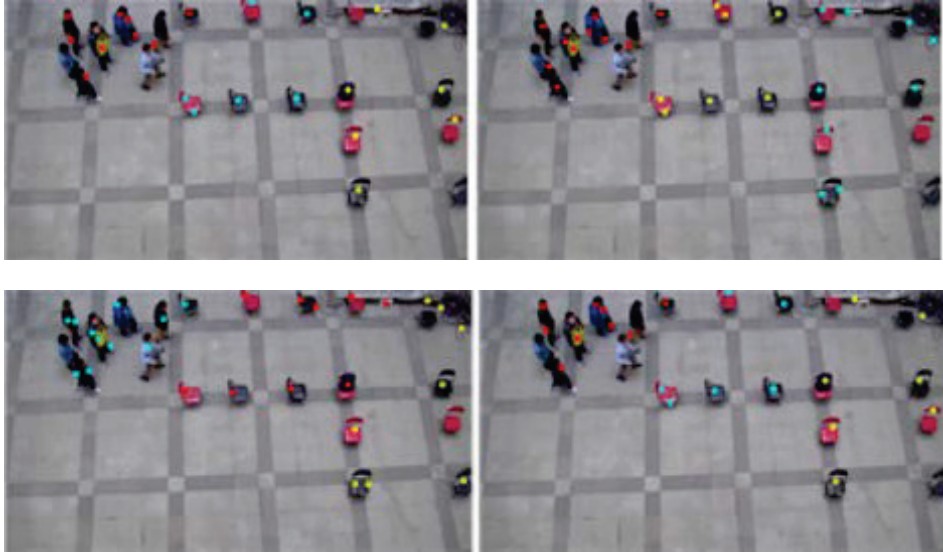

**Figure 5.** The clustering result with the 3-Component GMM of the 7-th, 8-th, 9-th, 10-th frames in My_video1.

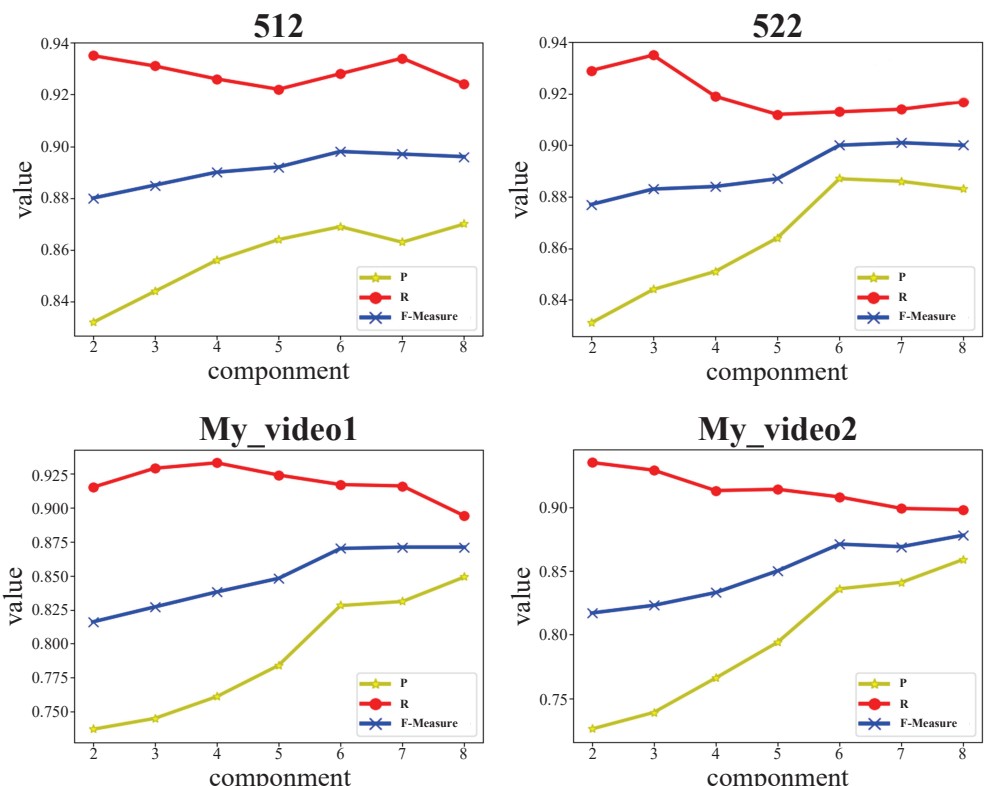

**Figure 6.** F, P, R of varying the number of GMM components.

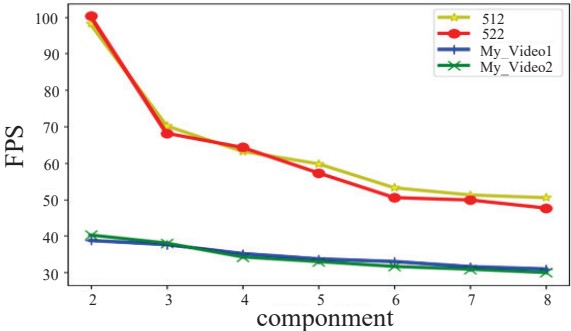

**Figure 7.** FPS of varying the number of GMM components.

3.3.3. Trajectory Based Class Classifier

Figure 5 shows that in the 1st, 2nd, and 4th clips, the red set represents the foreground, and the blue and yellow sets represent the background. In the 3rd clip, pedestrians were incorrectly detected as another cluster, unlike in the 1st, 2nd, 4th clip. The reason for this is that it is difficult to classify the foreground and background using a GMM because it is a clustering method.

Therefore, we propose a trajectory-based-classifier method to foreground and background. Suppose $G^{t-s+1} \cdots G^t$ are the clustering results for the first s frames:

$$G^t = \{g_1^t, g_2^t, \cdots, g_m^t\}, m = 1, 2, \cdots, k, \tag{7}$$

where $G^t$ means the $t$th frame clustering result and $m = k$ means there are $k$ cluster in $G^t$. $g_m^t = \{x_{m1}^t, x_{m2}^t, \cdots x_{mn}^t\}$, $n = 1, 2, \cdots$, num, where $g_m^t$ represents the $m$th cluster, $x_{nn}^t = (r_i^t, g_i^t, b_i^t, x_i^t, y_i^t)$ represents the pooled ROI central point, and num is the total number of ROI central points in $g_m^t$. Clusters in $G^t$ the current frame are matched one-by-one with clusters in of $G^{t-1}$ the previous frame and if the matching is successful, they are considered the same type of target. Our proposed trajectory-based-classifier method is as follows:

(1) Calculate the mean-feature $\bar{g}^t_m$ of each cluster

$$\bar{g}^t_m = \frac{1}{num} \sum_{n=1}^{mon} x^t_{mn} = \frac{1}{num} \left( \sum_{n=1}^{mm} r^t_n, \sum_{n=1}^{mm} g^t_n, \sum_{n=1}^{mun} b^t_n, \sum_{n=1}^{mon} x^t_n \sum_{n=1}^{mon} y^t_n \right). \tag{8}$$

The mean-feature of all clusters in each frame is:

$$\bar{G}^t = \left\{ \bar{g}^t_1, \bar{g}^t_2, \cdots, \bar{g}^t_m \right\}, m = 1, 2, \cdots, k. \tag{9}$$

(2) Find the same class in adjacent frames

In adjacent frames, if a cluster $g^{t-1}$ in the previous frame is the same type of target as a cluster $g^t$ in the current frame, they have similar $\bar{g}^t$. If they are not the same target type, they often have significant differences. The $\bar{g}^t$ of each cluster in the previous and current frames is ordered accordingly, from small to large. The $\bar{g}^t$ of the same sequence location is the same type of target because $\bar{g}^t$ of the same type of target is very similar in adjacent frames. In this way, we can compare two ascending-order clusters in adjacent frames to get the same classes of foreground objects, thus reducing the cost of computation. Then we use this method to match the same type of target in the first $s$ frames. For example, the first 4 frames matched the result shown in Figure 8. As a result, we changed the classification in Figure 5 and classified the pedestrians as red, the chairs in the middle as blue, and the chairs on the right as yellow, as shown in Figure 8.

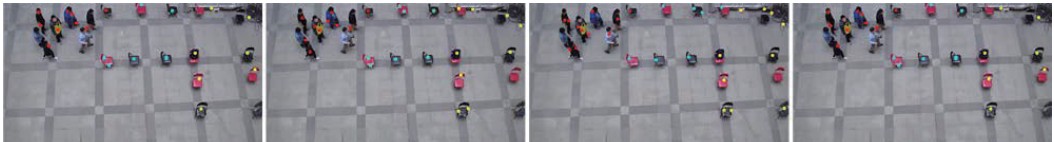

**Figure 8.** The first 4 frames matched this result.

(3) Label Positive and Negative Samples with Trajectories

We need to identify each class as foreground or background. When we obtain a class of targets in the first s frames, its displacement can be calculated. The proposed principle is as follows: The position coordinates of the mean-feature of the same cluster in the first s frames are denoted as $[(\bar{x}^{t-s+1}, \bar{y}^{t-s+1}), (\bar{x}^{t-s+2}, \bar{y}^{t-s+2}), \cdots, (\bar{x}^t, \bar{y}^t)]$. Then, we compute the moving distance of the clusters in $t-1, \ldots, t-s+2, t-s+1$ and $t$. We assume that the foreground displacements are increasing in a certain direction in adjacent frames, while the background often remains still or exhibits jitter with uncertain directions. If the moving distance of one cluster is increasing in the first s frames, it is classified as foreground; otherwise it is classified as background.

$$X = \begin{cases} 0, \sqrt{(\bar{x}^{t-1} - \bar{x}^t)^2 + (\bar{y}^{t-1} - \bar{y}^t)^2}, < \cdots, \\ < \sqrt{(\bar{x}^{t-s+2} - \bar{x}^t)^2 + (\bar{y}^{t-s+2} - \bar{y}^t)^2}, < \\ \sqrt{(\bar{x}^{t-s+1} - \bar{x}^t)^2 + (\bar{y}^{t-s+1} - \bar{y}^t)^2} \\ 1, \text{else} \end{cases} \tag{10}$$

where $X = 0$ means foreground, marked as $X_F$, and $X = 1$ means background, marked as $X_B$. After the foreground and background classes are categorized, the original samples can be used as a labeled training sample, represented as $X = \{X_F, X_B\}$, $v = \{0, 1\}$. In Figure 9, the red and yellow sets represent foreground and background, respectively. In this method, the correct selection of the value s has a great impact on the accuracy of the foreground and background detection. If s is too small, it is easy to misjudge a background element as foreground. If s is too large, the method may misjudge part of the foreground as

background. Therefore, we experiment on video test sets to decide $s$. We define precision for foreground and background detection as $c$:

$$c = \frac{sum(True)}{sum(Frame)},$$ (11)

where $sum(Frame)$ is the sum of the test frame sequences, and $sum(True)$ is the sum of the correctly judged test frame sequences. In Table 5, we find that when $s = 6$, c has the highest value in most cases, and can almost reach 0.9.

**Table 5.** The precision for classify the foreground and background.

| Dataset | Video Clips | 2 | 3 | 4 | 5 | 6 | 7 | 8 | 9 | 10 |
|---|---|---|---|---|---|---|---|---|---|---|
| **512** | 1499 | 0.528 | 0.683 | 0.856 | 0.914 | **0.924** | 0.919 | 0.903 | 0.839 | 0.789 |
| **522** | 1499 | 0.579 | 0.654 | 0.887 | **0.912** | 0.908 | 0.878 | 0.854 | 0.803 | 0.776 |
| **My_video1** | 390 | 0.471 | 0.589 | 0.718 | 0.842 | **0.903** | 0.879 | 0.803 | 0.753 | 0.684 |
| **My_video2** | 390 | 0.521 | 0.571 | 0.733 | 0.883 | **0.899** | 0.857 | 0.794 | 0.709 | 0.649 |

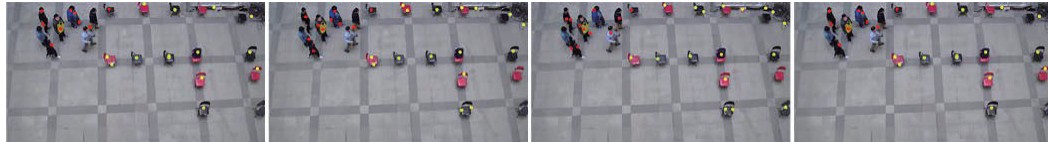

**Figure 9.** Result of detecting foreground and background: the red sets represent foreground and the yellow sets represent background.

*3.4. Online Sample Classifier*

3.4.1. Imbalance Compensation

The training samples are class imbalanced, as shown in Table 6. To better train the online classifier, we need to resample the training set to obtain balanced data. A K-means oversampling method is adopted to compensate for this imbalance. The methods are as follows:

(1) Calculate the total number of samples in the foreground, NF, and in the background, NB, respectively. Then, calculate the difference between them: $K = |NB - NF|$.

(2) Use the K-means method to preprocess minority classes to get K clusters, calculate the mean of each cluster, then use the means as new minority-class samples. The new samples are added to each training sample so we get a set of balanced training samples.

**Table 6.** Imbalance degrees of training sets.

| Imbalance Degree | 512 | 522 | Video1 | Video2 |
|---|---|---|---|---|
| $\eta$ | 0.184 | 0.538 | 0.421 | 0.523 |

The imbalance degrees are different in Tables 1 and 6, because we use ROI-based area filtering, mentioned in Section 3.2.

3.4.2. Online Sample Updating

We use the new balanced training samples as training sets for the Naive Bayesian classifier. We use the ROI center points of the latest frame as the test set. After the Naive Bayesian classifies the test set, we get foreground and background represented as follows:

$$X = \{[(r_1, g_1, b_1, x_1, y_1), \cdots (r_{N_F}, g_{N_F}, b_{N_F}, x_{N_F}, y_{N_F})],$$
$$[(r_1, g_1, b_1, x_1, y_1), \cdots (r_{N_B}, g_{N_B}, b_{N_B}, x_{N_B}, y_{N_B})]\}.$$

Thus, we can get background bounding boxes and copy the pixels in the bounding boxes to the previous frame. Using the frame difference algorithm to process the new previous frame and the current frame, the foreground can then be segmented. To update the trained samples online, the newly classified foreground and background are added to the training samples. Then, we train the Naive Bayesian classifier with the updated training samples so that we can detect the foreground online.

The experimental results of the training classifier with imbalanced training sets are shown in Figure 10, and the precision results are shown in Figure 11a. The experimental results of training the classifier with the balanced and updated training sets are shown in Figure 12, and the precision is shown in Figure 11b. Red represents the background and yellow represents the foreground, respectively. Figure 11a,b demonstrated that *P* increased by 22.7%, *R* increased by 2.1%, and *F* increased by 23.4%. This means using the proposed K-means oversampling method and online updating training samples to compensate for the weaknesses of the classifier is effective, especially for *P* and *F*.

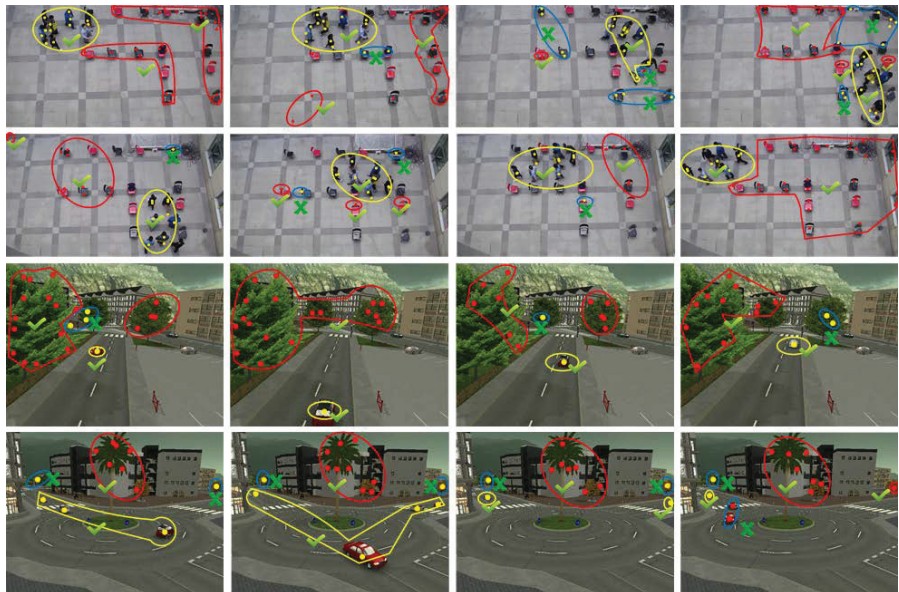

**Figure 10.** Experimental results from the Naive Bayes with imbalanced training sets. Top to bottom: results for "My_video1", "My_video2", "512", "522".

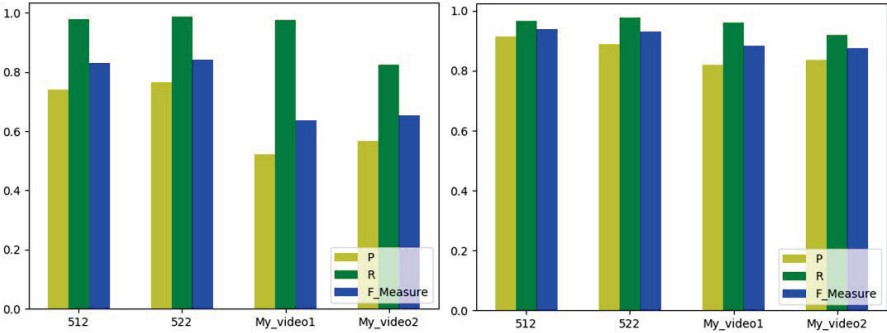

**Figure 11.** (**Left**) Performance evaluation for classifier with imbalanced training sets. (**Right**) Performance evaluation for classifier with the balanced and updated training sets.

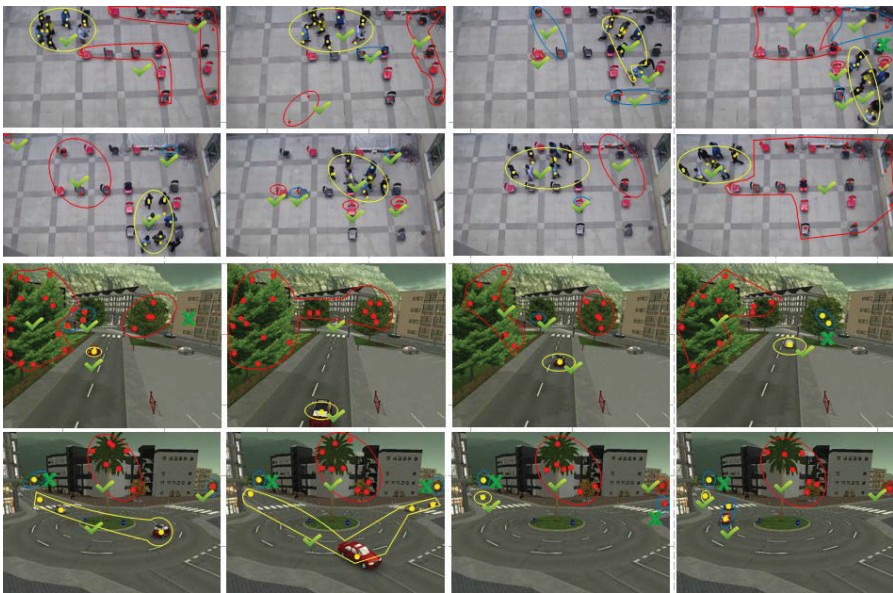

**Figure 12.** Experimental results from the Naive Bayes with the balanced and updated training sets. Top to bottom: results for "My_video1", "My_video2", "512", "522".

## 4. Evaluation

In this section, we compared RoiSeg with the traditional foreground segmentation algorithms. Because the CNN foreground segmentation algorithms are not suitable for these scenarios due to their strict real-time requirements. Sobral tested and compared 29 background subtraction algorithms and recommended five of the best, namely DP-WrenGABGS, MixtureOfGaussianV1BGS, MultiLayerBGS, PixelBasedAdaptiveSegmenter and LBAdaptiveSOM [38]. In this paper, we used these five algorithms to compare with our proposed method and the hardware of our experiment is a Lenovo desktop with Intel(R) Core(TM) i5-4590 CPU @ 3.3 GHz, 8 GB RAM, Win 10 64bit system. Because the foreground detected by frame difference has an aperture, we manually filled some aperture in the foreground in order to evaluate our algorithm using *P*, *R* and *F*. As the size of the self-captured sequences "My_video1" and "My_video2" were 1280 × 720 and that of the sequences provided by the BMC database was 640 × 480, the FPS we give was the average for all test sequences.

In Section 3.2, for the video sequences "112", "122", "212", "222", "312", "322", "412", and "422", we found that after filtering with the preset threshold, all foregrounds were recognized. Thus, we did not need a classifier to distinguish foreground and background. The results of running these five algorithms and the proposed RoiSeg on these eight sequences are shown in Table 4 and Figure 13. The average *P*, *R* and *F* of the five algorithms and proposed RoiSeg were computed, as shown in Table 4. We could see that the proposed RoiSeg had the best *R* and the best *F* on some sequences. For video sequences "112" and "122", which were without noise, and "212" and "222", which had salt and pepper noise, the proposed noise filter method did not function best; however, the *R* and *F* still reached over 0.9. For video sequences "312" and "322", which having moving cast shadows, and "412" and "422", which were foggy, the proposed noise filter method functions best. A non-optimal *P* indicated that our method produced more false positives than other algorithms. The reason for this was that the foreground detected by frame difference has ghosts, and the increased false positives were mainly located at the boundaries of moving objects, which was not harmful for visual observation. From Table 4, we found the proposed noise filter method had the highest FPS. This is due to the effectiveness of the proposed ROI-based noise filter, which mainly focuses on the ROI instead of the full frame.

For the video sequences "My_video1", "My_video2", "512", and "522", with wind and camera jitter, the noise could not be removed by filtering. Thus, we needed a classifier to distinguish foreground and background. The results of applying these five algorithms

and the proposed RoiSeg algorithm are shown in Table 4 and Figure 14. From Figure 14, we could observe that the proposed RoiSeg algorithm produced better visual results than the five algorithms. Furthermore, Table 4 shows that the RoiSeg algorithm had the best *F*, which proved that our method had the best overall performance. After the proposed clustering and online classification work in Sections 3.3 and 3.4, the FPS of our method decreases. However, Table 4 shows that the FPS of our method still outperformed those of the other approaches.

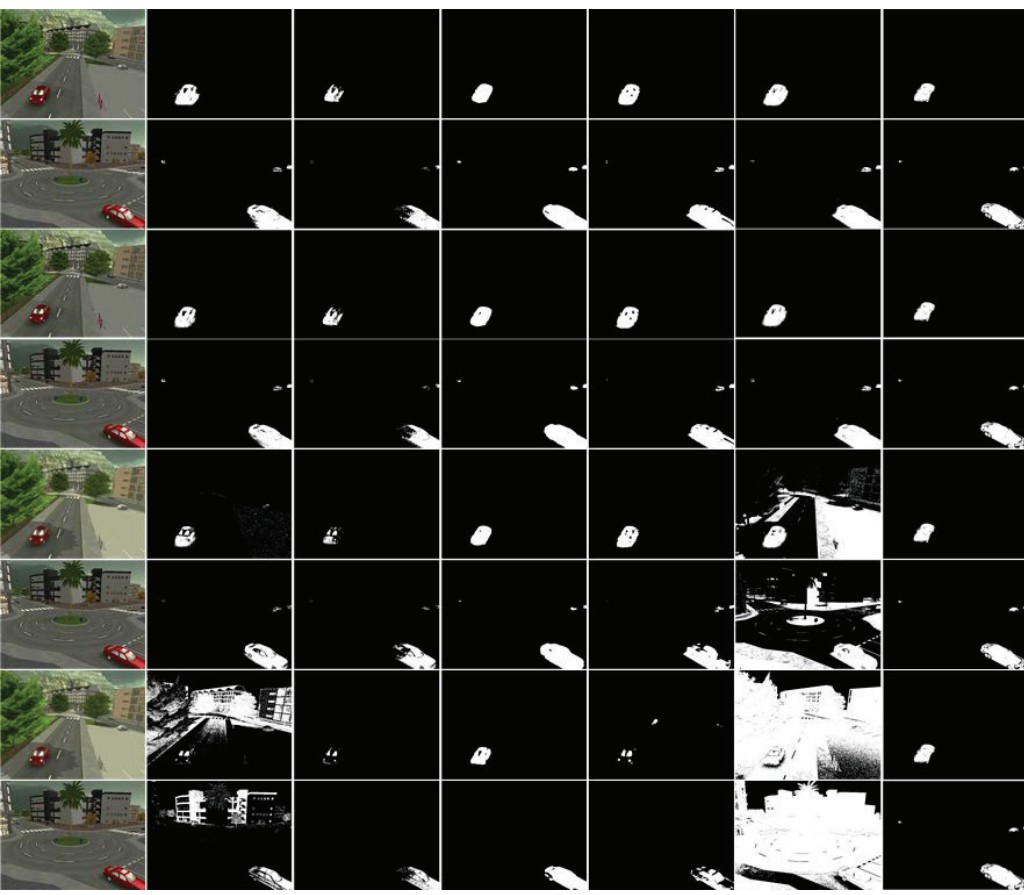

**Figure 13.** Experimental results on eight sequences. Left to right: original images, DPWrenGABGS, MixtureOfGaussianV1BGS, MultiLayerBGS, PixelBasedAdaptiveSegmenter, LBAdaptiveSOM, proposed method. Top to bottom: "112", "122", "212", "222", "312", "322", "412", "422".

We also evaluated the performance of RoiSeg on different datasets with the metrics of the average pixel error rate (APFPER) and the joint intersection overlap (IoU) [39]. APF-PER measured the number of misclassified pixels and IoU was to calculate the combined intersection of the estimated and true split plots for evaluating the split performance. We compared RoiSeg with state-of-the-art unsupervised learning methods on FBMS dataset, as shown in Figure 15 and Table 7. It is observed that the image saliency methods rendering of information within frames can produce unsatisfactory results, and even some images miss foreground objects, mainly because the time correlation in the image sequence to convey the target information was not taken into consideration [40]. However, these foreground segmentation methods based on motion perform better than the image saliency methods [41–43]. RoiSeg estimated the target object in a more cluttered background with higher real-time boundary and splits video objects in a completely unsupervised manner. We also conducted experiments to compare the performance of RoiSeg and other segmentation methods on a different dateset (SegTrack), as shown in Tables 8 and 9.

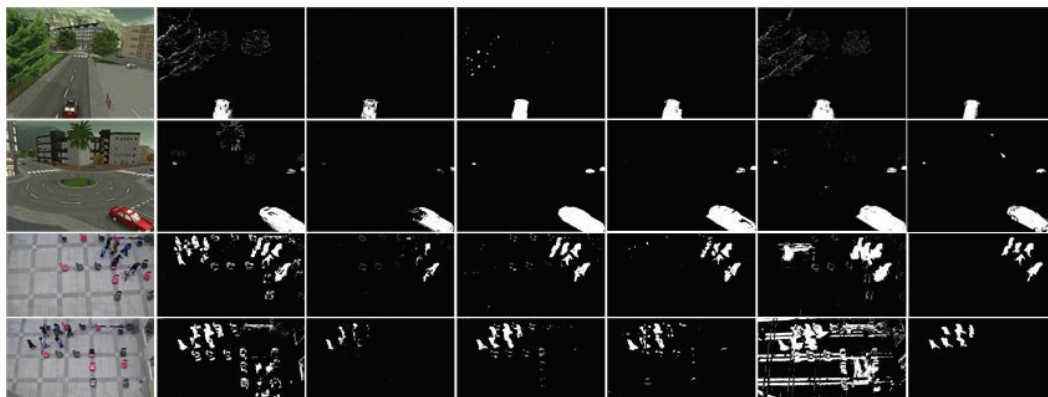

**Figure 14.** Experimental results on four sequences. Left to right: original images, DPWrenGABGS, MixtureOfGaussianV1BGS, MultiLayerBGS, PixelBasedAdaptiveSegmenter, LBAdaptiveSOM, proposed method. Top to bottom: "512", "522", "My_video1", "My_video2".

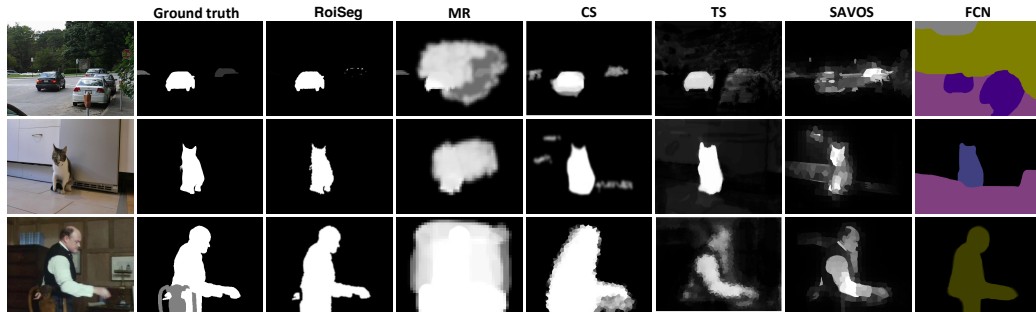

**Figure 15.** Comparison between RoiSeg and state-of-the-art unsupervised learning methods on FBMS.

**Table 7.** Comparison based on IoU between RoiSeg and other methods on FBMS dataset.

| Video | RoiSeg | [42] | [44] | [45] | [41] | [46] |
|---|---|---|---|---|---|---|
| Bear2 | 63.51 | 87.52 | 21.14 | 86.81 | 70.11 | 88.92 |
| Cars5 | 15.62 | 10.71 | 38.73 | 17.38 | 38.52 | 60.11 |
| Cars9 | 30.17 | 19.55 | 28.92 | 52.44 | 60.08 | 77.82 |
| Cats1 | 78.83 | 19.75 | 81.49 | 83.11 | 85.72 | 70.13 |
| People1 | 58.63 | 56.06 | 64.82 | 53.33 | 68.12 | 77.07 |
| People5 | 55.82 | 10.71 | 84.43 | 51.81 | 56.41 | 73.31 |
| Rabbits2 | 56.01 | 20.41 | 47.81 | 28.32 | 71.06 | 79.12 |
| Avg. | 51.23 | 32.10 | 52.48 | 53.31 | 64.29 | 75.21 |

In Table 8, References [41–43,45,47–50] are unsupervised learning methods, while [39,51] are the supervised learning methods. The results demonstrated that RoiSeg could meet the requirements of most tasks, although its performance was not as good as state-of-the-art segmentation methods. In Table 9, References [41,44,45,48,52] are unsupervised learning methods, while [46,50,51] are supervised learning methods. Among them, ref [46] utilized the CNN methods and VOC 2011 [53] for pre-training and testing, respectively. Table 9 shows that the result of IoU evaluation on RoiSeg was similar to that of APFPER and the CNN-based approaches had an absolute advantage in supervised segmentation tasks, but it relied on too much data. We also conducted extensive experiments to evaluate the real-time performance as shown in Table 10 and RoiSeg achieved a better performance in terms of real-time operations. In summary, RoiSeg outputted the expected results on some video sequences compared to the best performing unsupervised learning methods. There is a

gap compared to the supervised method and the CNN method, and RoiSeg is better in real-time operations with the average processing time of 45 ms.

**Table 8.** Comparison based on APFPER between RoiSeg and other methods on SegTrack dataset.

| Video | Frames | Unsupervised | | | | | | | Supervised | | |
|---|---|---|---|---|---|---|---|---|---|---|---|
| | | RoiSeg | [47] | [44] | [45] | [48] | [49] | [50] | [41] | [39] | [51] |
| Birdfall | 30 | 352 | 217 | 155 | 189 | 144 | 199 | 468 | 140 | 252 | 454 |
| Cheetah | 29 | 776 | 890 | 633 | 806 | 617 | 599 | 1968 | 622 | 1142 | 1217 |
| Girl | 21 | 1253 | 3859 | 1488 | 1698 | 1195 | 1164 | 7595 | 991 | 1304 | 1755 |
| Monkeydog | 71 | 557 | 284 | 365 | 472 | 354 | 322 | 1434 | 350 | 563 | 683 |
| Parachute | 51 | 412 | 855 | 220 | 221 | 200 | 242 | 1113 | 195 | 235 | 502 |
| Avg. | | 670 | 1221 | 572 | 677 | 502 | 505 | 2516 | 459 | 699 | 922 |

**Table 9.** Comparison based on IoU between RoiSeg and other methods on SegTrack dataset.

| Video | Frames | Unsupervised | | | | | | Supervised | | |
|---|---|---|---|---|---|---|---|---|---|---|
| | | RoiSeg | [44] | [45] | [52] | [48] | [41] | [50] | [51] | [46] |
| Birdfall | 30 | 60.91 | 71.43 | 37.39 | 72.52 | 73.21 | 74.51 | 78.71 | 57.41 | 78.83 |
| Cheetah | 29 | 50.12 | 58.75 | 40.91 | 61.21 | 64.22 | 64.34 | 66.12 | 33.82 | 75.31 |
| Girl | 21 | 70.94 | 81.91 | 71.21 | 86.37 | 86.67 | 88.72 | 84.64 | 87.85 | 88.84 |
| Monkeydog | 71 | 65.21 | 74.24 | 73.58 | 74.07 | 76.12 | 78.04 | 82.15 | 54.35 | 85.65 |
| Parachute | 51 | 90.12 | 93.93 | 88.08 | 95.92 | 94.62 | 94.8 | 94.42 | 94.52 | 95.61 |
| Avg. | | 67.46 | 76.05 | 62.23 | 78.02 | 78.97 | 80.08 | 81.21 | 65.59 | 84.85 |

**Table 10.** Realtime comparison between RoiSeg and other methods.

| Method | RoiSeg | [44] | [45] | [50] | [51] | [41] | [46] |
|---|---|---|---|---|---|---|---|
| Value | 0.05 | 35.12 | 0.53 | 1.82 | 0.84 | 3.26 | 0.38 |

## 5. Conclusions and Future Work

In this paper, we propose RoiSeg, an effective object segmentation method, which consists of three modules, ROI-central-point generation and feature extraction, automatic training-sample generation, and an online sample classifier. RoiSeg can be applied to a number of scenarios where datasets are difficult to obtain and require high real-time performance. We also conduct extensive experiments and the results demonstrate that the frames per second of RoiSeg is 95.84, which is better than other algorithms, and the classification accuracy is 92.4%. Future work may fall into two categories. First, to find better algorithms to detect stopped objects, we plan to introduce Kalman filtering to predict the state of the stopping target the next time. Second, we will try to design a deep neural network algorithm to study the segmentation of the foreground in long-term scenarios.

**Author Contributions:** Conceptualization, Z.Z. and Z.P.; data collection, Z.T.; analysis and interpretation of results, Z.Z. and F.G.; validation, Z.Z. and F.G.; writing—original draft preparation, Z.Z. and F.G.; writing—review and editing, F.G. All authors have read and agreed to the published version of the manuscript.

**Funding:** This work was supported by China Postdoctoral Science Foundation (2020M671597), Jiangsu Postdoctoral Research Foundation (2020Z100), the National Science Foundation of the Jiangsu Higher Education Institutions of China (20KJB520002), Suzhou Planning Project of Science and Technology (No. SYG202024), and the Priority Academic Program Development of Jiangsu Higher Education Institutions (PAPD).

**Institutional Review Board Statement:** Not applicable.

**Informed Consent Statement:** Not applicable.

**Data Availability Statement:** BMC (Background Models Challenge) provides videos for testing our background subtraction algorithm. For more description of the BMC dateset and how to use the BMC dateset, please refer to this website: http://backgroundmodelschallenge.eu/#evaluation; (accessed on 2 January 2022). The Freiburg-Berkeley Motion Segmentation Dataset (FBMS) has a total of 720 frames annotated. FBMS-59 comes with a split into a training set and a test set. For more description of motion segmentation dataset and how to use evaluation code, please refer to this website: https://lmb.informatik.uni-freiburg.de/resources/datasets/moseg.en.html; (accessed on 2 January 2022). SegTrack is a video segmentation dataset. For more description of SegTrack dataset, please refer to this website: https://web.engr.oregonstate.edu/~lif/SegTrack2/dataset.html; (accessed on 2 January 2022).

**Conflicts of Interest:** The authors declare no conflict of interest.

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
