# Peer review of "RoiSeg: An Effective Moving Object Segmentation Approach Based on Region-of-Interest with Unsupervised Learning"

_applsci, doi:10.3390/app12052674_

Round 1

Reviewer 1 Report

The authors developed a method to segment moving objects considering a region of interest in the central region of the frames and unsupervised learning. The proposed method can handle some problems found in the images and has a better performance than other segmentation algorithms do. There are, however, some issues to be improved in the manuscript.

1) When presenting and discussing the results, the authors should use the past verb tense. I suggest reviewing the writing to correct such an issue in the manuscript.

2) Referring to figures with just their numbers is unusual and leads to confusion. I recommend adding "Figure" or "Fig." before their numbering to improve the readability of the manuscript.

Author Response

We really appreciate you for all your comments to improve the quality of our paper. We have revised the paper in line according to your comments.

Comment 1.1

1) When presenting and discussing the results, the authors should use the past verb tense. I suggest reviewing the writing to correct such an issue in the manuscript.

Response 1.1

Thanks for your suggestions. We have used the past verb tense in presenting and discussing the results in our revised manuscript.

Comment 1.2

Referring to figures with just their numbers is unusual and leads to confusion. I recommend adding "Figure" or "Fig." before their numbering to improve the readability of the manuscript.

Response 1.2

Thanks for your reminding. We have added "Fig." in our revised manuscript.

Reviewer 2 Report

Authors proposed ROI-central-point generation and feature extraction, automatic training-sample generation, and classification. Literature review and background research is enough to understand fundamental research about related work. English grammar looks fine. Simulated results for classification is interesting. Therefore, the manuscript can be minor revision. However, authors need to revise the manuscript according to the comments as below. 

  1. Figure 1 fonts are small to be seen.
  2. Figure 2 label size need to be increased.
  3. Figures 3, 4, and 5 had better combined to one Figure.
  4. Figures 6 and 7 had better combined to one Figure.
  5. Figures 9 and 10 labels are too small.
  6. Tables 6,7, and 8 had better combined to one Table.
  7. Authors need to have abbreviated journal names in the reference sections.
  8. No data availability section, No funding, and No acknowledgement sections.
  9. In the conclusion section, authors need to emphasize important data.
  10. In Line 366, please change [41-48] for the references.

Author Response

We would like to thank you for reviewing our paper and for constructive advice which helps us improve the quality of our paper. We have revised the paper in line according to your comments.

Comment 2.1

Figure 1 fonts are small to be seen.

Response 2.1

Thanks for your suggestions. We have updated the Figure 1 in our revised manuscript.

Comment 2.2

Figure 2 label size need to be increased.

Response 2.2

Thanks for your suggestions. We have updated the Figure 2 in our revised manuscript.

Comment 2.3

Figures 3, 4, and 5 had better combined to one Figure.

Response 2.3

Thanks for your suggestions. We have combined the original Figure 3, 4, and 5 into one figure in our revised manuscript.

Comment 2.4

Figures 6 and 7 had better combined to one Figure.

Response 2.4

Thanks for your suggestions. We have combined the original Figure 6 and 7 into one figure in our revised manuscript.

Comment 2.5

Figures 9 and 10 labels are too small.

Response 2.5

Thanks for your suggestions. We have updated the labels of these two figures in our revised manuscript.

Comment 2.6

Tables 6,7, and 8 had better combined to one Table.

Response 2.6

Thanks for your suggestions. We have combined the original Tables 6, 7, 8, and 9 into one table in our revised manuscript.

Comment 2.7

Authors need to have abbreviated journal names in the reference sections.

Response 2.7

Thanks for your suggestions. We have corrected these errors in our revised manuscript.

Comment 2.8

No data availability section, No funding, and No acknowledgement sections.

Response 2.8

Thanks for your suggestions. We have added these sections in our revised manuscript.

Comment 2.9

In the conclusion section, authors need to emphasize important data.

Response 2.9

Thanks for your suggestions. We have emphasized the important data in our conclusion section.

Comment 2.10

In Line 366, please change [41-48] for the references.

Response 2.10

Thanks for your suggestions. We have changed the original format into [41-48] in our revised manuscript.

Reviewer 3 Report

The paper is well-written and can be accepted. I just have one comment. The subscript "i" that has been introduced in Section 3.3.1 has not been defined anywhere in the paper.

Author Response

We really appreciate you for all your comments to improve the quality of our paper. We have revised the paper in line according to your comments.

Comment 3.1

The paper is well-written and can be accepted. I just have one comment. The subscript "i" that has been introduced in Section 3.3.1 has not been defined anywhere in the paper.

Response 3.1

Thanks very much for your review. We have added the explanation of "i" in our revised manuscript.